# The Comparative Efficacy of Treatments for Children and Young Adults with Internet Addiction/Internet Gaming Disorder: An Updated Meta-Analysis

**DOI:** 10.3390/ijerph19052612

**Published:** 2022-02-24

**Authors:** Chuan-Hsin Chang, Yue-Cune Chang, Luke Yang, Ruu-Fen Tzang

**Affiliations:** 1Genomics Research Center, Academia Sinica, Taipei 115, Taiwan; chuanhsin032484@gmail.com; 2Department of Mathematics, Tamkang University, New Taipei City 251301, Taiwan; ycchang414@gmail.com; 3Department of Social Welfare, Hsuan Chuang University, Hsinchu 30092, Taiwan; LHY@hcu.edu.tw; 4Department of Psychiatry, Mackay Memorial Hospital, Taipei 10449, Taiwan; 5Department of Childhood Care and Education, Mackay Junior College of Medicine, Nursing, and Management, Taipei 25245, Taiwan; 6Department of Audiology and Speech-Language Pathology, Mackay Medical College, New Taipei City 251301, Taiwan

**Keywords:** internet addiction, internet gaming disorder, psychological treatment, pharmacotherapy, meta-regression

## Abstract

Internet gaming disorder (IGD) is a formal mental disorder leading to bad outcomes for children and adolescents. This study comprehensively compared the estimated effect of various pharmacotherapy and psychosocial interventions for IGD from randomized controlled trials (RCT) through updated meta-analysis, using meta-regression. A search of PubMed/MEDLINE, Cochrane Library, and Airiti Library between 2000 and 2017 was conducted for various IA/IGD intervention modalities. A total of 124 studies from 29 selected papers involving 5601 children and young adults with IA/IGD were found. Meta-analyzing the pooled standardized mean difference (SMD) revealed a preliminary random effect of 1.399 with a 95% confidence interval of 1.272–1.527, suggesting highly effective treatment of IA/IGD. After adjusting for the confounding risks of age, publication year, type of subjects, and type of study, this study revealed that combining pharmacotherapy with cognitive behavioral therapy (CBT) or multi-level counseling (MLC) was the most effective treatment option. Using a scale of time spent online or a severity of IA symptoms scale was a more effective measurement, with *p*-values = 0.006 and 0.002, respectively. IA/IGD patients with comorbid depression showed worse outcomes than youth with another comorbidity. The corresponding model goodness-of-fit indices were τ^2^ = 1.188; I^2^-Residual = 89.74%; and Adjusted-R^2^ = 16.10%. This systematic review indicates that pharmacotherapy combined with CBT or MLC might be an effective therapeutic strategy for youth with gaming disorder.

## 1. Introduction

Children and adolescents have had an increasing risk of internet addiction (IA) or internet gaming disorder (IGD) during the COVID-19 pandemic. Generally, the prolonged use of internet and gaming due to social restrictions related to the COVID-19 pandemic or limited human face-to-face interpersonal contact during the COVID-19 pandemic has resulted in an increasing rate of internet addiction or gaming disorder [1,2,3]. However, worldwide mental health experts have been more concerned about depression and anxiety symptoms among children and adolescents during the COVID-19 pandemic [4]. In fact, mental health professionals might also worry more about the psychological and physical consequences caused by internet gaming disorder, and increasingly focus on developing treatment and preventative intervention for IGD [5].

Internet addiction (IA) or internet Gaming Disorder (IGD) was rising significantly in prevalence among children and adolescents before the COVID-19 pandemic. A dramatically increased number of youths from all over world have become accustomed to having access to smart phones and wireless internet [6]. The prevalence of pathological gaming in the United States and Europe is rising and ranges between 1.5% to 8.2% among the general population [7]. In the child and adolescent population, internet gaming disorder prevalence as high as 8% has been observed, as well [8,9,10]. Indeed, gaming disorder inevitably influences children’s mental health world-wide, with prevalence rates of 2.1 to 2.6% of youth in North America, 0.2 to 4.4% in Oceania, and 0.2 to 12.3% of youth in Europe [11]. Internet gaming disorder is rising significantly among child and adolescent populations.

Past research has found that internet gaming disorder might contribute to the following bad mental health consequences for youth: academic decline, sleep deprivation [12], interpersonal relationship problems, family conflict [13], physical health problems, loneliness, suicidal ideation [14], aggression, depression [15], social withdrawal, dissociative experiences [16], cybercrimes [17], and future decline in workplace competitiveness [18]. Therefore, as of 2018, the World Health Organization formally enrolled internet gaming disorder as a mental disorder. Accordingly, there is a strong need for diagnosing IGD earlier and providing effective early treatment strategies for gaming-addicted youth. Moreover, an updated meta-analysis of treatment efficacy for gaming disorder is essential for clinicians’ decision making.

According to Winkler et al. in 2013, effective treatment options for internet addiction include combining pharmacological therapies with psychological intervention (with Hedges’ *g* = 1.61), based on a meta-analysis of randomized control trials [19]. Internet gaming disorder is often comorbid with other childhood neurodevelopmental disorders such as Attention Deficit Hyperactivity Disorder (ADHD) [20], depression, or anxiety [15]. Therefore, using pharmacotherapy for youth with internet gaming disorder is increasingly important. For example, different antidepressant drugs such as escitalopram (for depression) and bupropion (for impulsivity) [21] and psychostimulant drugs such as methylphenidate have all been reported as effective for internet gaming disorder [22]. Aside from pharmacotherapy for IGD, many psychosocial approaches are available as interventions for internet gaming disorder. Many psychosocial interventions for internet gaming disorder, including cognitive behavioral therapy (CBT) [23], acceptance and commitment therapy (ACT), reality therapy (RT), and multi-level counseling (MLC) programs have become rising stars in this field. The forms of psychological therapy intervention for youth with IA/IGD were delivered by individual-session psychotherapy, family-based therapy, and group-based supportive psychotherapy.

However, in 2017, both Zajac et al. and King et al. showed concerns about the quality of studies on interventions for internet gaming disorder in the past 10 years [24,25]. They recommended a more well-established or updated meta-analysis on treatment outcomes for internet gaming disorder [24,25,26]. Indeed, in 2013, Winkler et al. found it difficult to estimate the effect of internet gaming disorder more thoroughly, possibly due to a small sample size and insufficient availability of literature at that time [19]. As a result, until now, the long-term efficacy of combination therapy for IGD in youth still remains unknown. In addition, the effect estimate can also be influenced by measurement tool use [27]. Widely used international screening tools, including the Internet Addiction Test (IAT) proposed by Young et al. in 1998 [28], the Compulsive Internet Use Scale developed by Meerkerk et al. [29], and the Problematic Internet Use Questionnaire devised by Demetrovics et al. [30] were definitely influencing the effect size (ES) of internet addiction. We systematically reviewed a total of 45 existing internet addiction scales in 23 languages; researchers had concluded before that only 17 of the scales’ psychometric properties had been evaluated [31]. In addition, the treatment effect of IA/IGD can be influenced by confounding variables such as age, type of treatment, scales for treatment effect, types of comorbid diagnosis, and types of study. Accordingly, there is a need to use an updated meta-analysis to systemically review the effect size (ES) of internet addiction by using different IA/IGD screening measurements for children and young adults. Therefore, we have strived to collect more randomized clinical trials, including those of various recent pharmacotherapy and psychosocial interventions for IA/IGD from Western and Eastern countries, to determine the best intervention for internet gaming disorder, with an updated meta- analysis using meta-regression.

Accordingly, in this study, we first compare the effect size of pharmacotherapy and psychosocial treatments of IGD with an updated meta-analysis. We will explore meta-regression to address whether the aforementioned potentially confounding risk factors, e.g., age, type of treatment, scales for treatment effect, types of comorbid diagnosis, types of study-designed data, group therapy (Yes vs. No), and parental involvement in the therapy program (Yes vs. No), can influence the treatment effects of IA/IGD in children and young adults. Such a systematic review using meta-regressions of randomized controlled trials is a vital way to provide clinicians with the confidence to choose the right treatment option for internet gaming disorder in youth.

## 2. Methods

### 2.1. Search and Study Selection

We searched four databases (PubMed, Cochrane Library, SCOPUS, and Airiti Library) for articles published between 2000 and 2017. Two independent scientists (Chang and Tzang) screened the literature and identified a meta-analysis of randomized control trial studies on effective treatments for IA/IGD in children and young adults.

The following full electronic search terms were used: (“Internet Addiction” OR “Internet Gaming Disorder” OR “Problematic Internet Use” OR “Internet Usage” OR “Internet Overuse” OR “Problematic Internet” OR “Online Addiction” OR “Cybersex” OR “Computer Game Addiction”) AND (treatment OR therapy OR program OR training OR psychotherapy OR curriculum OR psychosocial OR Escitalopram OR Methylphenidate OR Bupropion OR Atomoxetine OR Fluoxetine OR pharmacological OR non-pharmacological OR behavior OR Treat * OR Therapy * OR Intervention OR Psychotherapy) AND (child OR “elementary students” OR adolescent OR “Young Adult”). To avoid overlooking any crucial studies, we also searched reference lists of the retrieved articles to identify other eligible studies.

### 2.2. Inclusion and Exclusion Criteria

Studies were included if they met the following inclusion criteria: (1) the study addressed the “internet addiction” topic (or a related topic, e.g., internet gaming disorder); (2) the age of all participants was less than 25 years (intended to include college students); (3) the study included a pharmacological or non-pharmacological treatment (mainly psychological without excluding others, e.g., alternative medicines or therapies) aimed to decrease IA-related problems (mainly, severity of IA, time spent online, depression, and anxiety).

Studies meeting the following criteria were excluded: (1) the study did not provide sufficient information to evaluate the (Hedges’ *g*) effect sizes, and the authors did not respond to our mail or were not willing to provide additional information; (2) the study did not report results for at least one of the aforementioned four outcome variables; (3) the study was a case study; (4) unavailability of the full text of the study; and (5) a study was repeated, the participants overlapped either partially or completely with the subjects of another study, or a study was included in a meta-analysis. There were 124 eligible studies included in this study after application of the inclusion and exclusion criteria.

### 2.3. Data Extraction

The following were all variables for treatment efficacy for gaming disorder: (1) the last name of the first author; (2) publication year; (3) type of treatment; (4) treatment duration (in weeks); (5) number of subjects in the analysis; (6) mean age of subjects; (7) type of comorbid diagnosis (IA, IA + ADHD, and IA + depression); (8) type of scales used to evaluate the treatment effects; (9) group therapy (Yes vs. No); and (10) parents involved in the therapy program (Yes vs. No).

To explore potential influential factors and reduce heterogeneity, we recoded the type of treatment, the type of scales used to evaluate the treatment effects, the type of comorbid diagnosis, and the types of study design as following.

#### 2.3.1. Types of Treatment

Interventions were classified as the following five types of treatment: (1) Treatment 1: Drugs (methylphenidate, escitalopram, bupropion, atomoxetine, and fluoxetine); (2) Treatment 2: Cognitive behavioral therapy (CBT), positive psychology interventions, psychological intervention, traditional psychology intervention, and integrated internet addiction prevention program); (3) Treatment 3: multi-level counseling program (MLC), solution-focused brief therapy (SFBT) with family therapy, Adlerian group counseling, multi-family group therapy, solution-focused group counseling, interpersonal group program; (4) Treatment 4: Drugs + CBT or MLC; (5) Treatment 5: Others + CBT or MLC; (6) Treatment 6: Others (electro-acupuncture (EA), psychological nursing intervention, sports, negative health effect intervention, Adlerian group counseling, electroencephalographic (EEG) biofeedback treatment, Han’s acupoint nerve stimulation (HANS), and knowledge, Attitude, belief, and practice (KABP)).

#### 2.3.2. Types of Symptom Measurement Scales

The types of scales used to evaluate the treatment effects were classified as follows: (1) Type 1: Anxiety (Hamilton Anxiety Scale (HAMA), Screen for Child Anxiety Related Emotional Disorders (SCARED), Self-Rating Anxiety Scale (SAS), SCL-90Anxiety score, State-Trait Anxiety Inventory (STAI)); (2) Type 2: Depression (Hamilton Depression Scale (HAMD), Hamilton Depression Rating Scales (HDRS), Self-Rating Depression Scale (SDS), SCL-90Depression score; Children’s Depression Inventory (CDI)); (3) Type 3: Severity of internet addiction (Chen/Chinese Internet Addiction Scale (CIAS), Chinese Internet Addiction Scale Revised (CIAS-R), Computer Gaming Addiction Intervention (CGAI), Checklist for the Assessment of Internet and Computer Game Addiction (AICA-C), Impulsive-Compulsive Internet Use Disorder version of the Yate-Brown Obsessive Compulsive Scale (IC-IUD-YBOCS), Internet Addiction Disorder (IAD) self-rating scale (ISS), Internet Addiction Disorder Diagnosis Questionnaire (IAD-DQ), Young’s Diagnostic Questionnaire for internet addiction (YDQ), Young’s Internet Addiction Scale (YIAS), YIAS-Korean version (YIAS-K), Internet Overuse Self-Rating Scale (IOSRS), Callan Internet Use Scale (CIUS), Self-Rating Internet Use Scale (SIUS), Internet Addiction Test (IAT), Adolescent Pathological Internet Use Scale (APIUS), IA Knowledge, Attitude, and Practice (KAP); (4) Type 4: Scales of time spent o (Time Management Disposition Scale (TMDS), severity of internet use (hours of internet use, SI)); (5) Type 5: Others (SCL-90 (Total), Strength and Difficulties Questionnaire (SDQ), negative health effects (NHE)).

#### 2.3.3. Type of Comorbid Diagnosis

The types of comorbid diagnosis were classified as follows: (1) PType 1: IA; (2) PType 2: IA comorbid with depression (IA + Depression); (3) PType 3: IA comorbid with ADHD (IA + ADHD).

#### 2.3.4. Types of Study Design

The term ”effect size” is frequently used in the social sciences, particularly in the context of meta-analysis. Effect sizes typically refer to versions of the standardized mean difference (SMD) as Hedges’ (adjusted) *g*. However, the value of the SMD is influenced significantly by the choice of control group, mainly depending on whether the study used an active control (e.g., drug (methylphenidate) vs. drug (atomoxetine)) or a placebo control (e.g., drug (methylphenidate) vs. placebo). The value of the SMD for the latter should be significantly higher than the value for the former. To be comparable and consistent, in this study, we classified types of study design as follows: (1) Study Type 1: For placebo control studies with an available mean pre–post change, the SMD of Hedges’ g was calculated using the mean pre–post change in both treatment and control groups. For comparison to active control studies, we treated this type of study as two independent groups with a mean pre–post change available and calculated each group’s unbiased SMD of Hedges’ g according to the following Study Type 2; (2) Study Type 2: For convenience, we first used 0.7 as the correlation coefficient between pre- and post-treatment scores to estimate the standard errors of pre- and -post changes for one group. Next, we used the mean pre–post change and the standard errors of the changes to calculate the unbiased SMD of Hedges’ *g*; (3) Study Type 3: This type consisted of two groups with only mean post-treatment scores available, that is, the pre-test information of both groups was not presented in the study. The unbiased SMD of Hedges’ *g* was calculated using the post-test means and standard deviations from both treatment and control groups; (4) Study Type 4: At times, a meta-analysis paper only presented an unbiased SMD of Hedge’s *g* and its standard errors. In that case, we used both pieces of information directly to analyze.

## 3. Statistical Analysis

The computer application used to analyze all data was STATA/SE version V13.0 for WINDOWS software (StataCorp. 2013. Stata Statistical Software: Release 13. College Station, TX, USA: StataCorp LP). We first used a fixed-effect meta-analysis to evaluate pooled effects sizes. A random-effects meta-analysis was performed when the studies’ heterogeneity was highly significant.

However, the SMDs of Hedges’ g were expected to be comparable, but not identical across studies due to highly significant heterogeneity among them (e.g., six types of treatment and five types of IA scales). For example, there were five types of scales used to evaluate the effect size, and as shown in the previous section, each scale type included many different formats for the same evaluation purpose. It should be noted that the SMD method does not correct for differences in the direction of the scale. Among all evaluation scales, there were mainly two different kinds of trends to represent the treatment effects, that is, either the higher value represents the better treatment effect or, from the reverse direction, the worse effect. For some scales, it is essential to multiply the mean values from one set of studies by −1 (or alternatively, to subtract the mean from the maximum possible value for the scale) to ensure that all the scales point in the same direction. To be consistent and to facilitate the interpretation of the results, we rearranged all the values of SMDs of Hedges’ (adjusted) *g* to be in one direction according to the characteristics of their original scales, that is, a larger effect size indicates a better effect estimate.

## 4. Additional Analysis

Meta-regression is an extension to subgroup analyses that allows the effect of continuous, as well as categorical, characteristics to be investigated, and in principle allows the effects of multiple factors to be investigated simultaneously. Therefore, meta-regression can use even the continuous, as well as categorical symptom scales, to investigate unbiased efficacy. Meta-regression is an effective tool for exploratory analyses of heterogeneity and for studies of cross-level interactions [32]. Heterogeneity can be reduced through interaction analysis (stratified or sub-group analysis) [33]. Meta-regression can merge meta-analysis and linear regression principles to better clarify the linear relationship of various outcome measures, and thereby provide clinicians and healthcare decision makers with more valuable information than that from a meta-analysis [34].

Meta-regression is appropriate for testing whether there are certain confounders of the outcome and whether selected variables (the independent variables) confound the effect size (the dependent variable). In this study, the influence of the meta-regression analyses was used to compare the potential factors influencing treatment effects. The dependent variable was the SMD of Hedges’ (adjusted) *g* and was assessed by the aforementioned four types of study-designed data. The independent variables were publication year, treatment duration, mean age, the aforementioned recoded type of treatment, scales for treatment effect, types of comorbid diagnosis, types of study-designed data, group therapy (Yes vs. No), and parents involved in the therapy program (Yes vs. No). The risk of bias was evaluated by funnel plots and Egger’s regression intercept tests for bias or Begg’s test. The level of significance was set at *p* < 0.05.

## 5. Results

In total, the meta-analysis consisted of 124 studies of 5601 subjects with IA/IGD (or comorbid with depression or ADHD), extracted from 29 published papers on the effects of various interventions (Table 1). Two of them were conducted in Taiwan, four were conducted in the USA, and one was conducted in Iran. The remaining papers were published by researchers from mainland China or Korea. A detailed summary of the characteristics of all the included clinical trials is presented in Table 1. A list of the 29 published papers with the names of the first authors and their titles is shown in Appendix A.

Using the SMD as the pre-specified effect size, the fixed-effect meta-analysis showed that there was a positive and effective overall pooled effect with highly significant heterogeneity (chi-squared = 2951.24, d.f. = 206, *p* < 0.001). Accordingly, the random-effect meta-analysis followed, and the pooled overall effect = 1.399 (with a 95% confidence interval of 1.272, 1.527). The level of evidence in the included studies showed the variation in ES attributable to heterogeneity; I-squared was 93.0%, and the estimate of between-study variance, Tau-squared, was 0.7492.

### 5.1. Univariate Meta-Regression Analysis

A meta-regression analysis was used to explore potential factors influencing the high heterogeneity of the ES. In Table 2, the results of univariate meta-regression showed that: ignoring other potential influencing factors’ effects, (1) the treatment effect on IA/IGD significantly increased while age increased (*p* = 0.013) (age effect); (2) pharmacotherapy combined with cognitive-behavioral therapy or MLC (Treatment 4) was the best treatment type among the five types of treatment modalities for IA/IGD (treatment effect); (3) the treatment effect was better while using a measurement of time spent online (Type 4) followed by the severity of internet addiction scale (Type 3) (measurement scale effect); (4) publication year, types of SMD, types of diagnosis, group therapy (Yes vs. No), and whether parents were involved in the therapy program (Yes vs. No) had no significant impact on the treatment effects.

### 5.2. Multiple Meta-Regression Analysis

We further used the multiple meta-regression models to explore the confounding risks on the effect estimate after mutually adjusting for the effects of other risk factors in the model. The merit of this study was that we rearranged all the values of SMDs of Hedges’ *g* to be in one direction according to the characteristics of their original scales, that is, a larger value indicates a better treatment effect. This way, the results of the meta-regression model became interpretable. As shown in Table 3, after adjusting for risk of the effects of treatment type and measurement scales on the effect estimate, the effect size significantly increased (on average) by 8.7% for each increasing year of the mean age of IA/IGD youth (*p* = 0.001). Similarly, after adjusting for the effects of mean age and type of measurement scales, the effect estimate was significantly higher with the use of pharmacotherapy + CBT or MLC (Treatment 4), Others + CBT or MLC (Treatment 5), and CBT (Treatment 2) than with the use of pharmacotherapy alone (Treatment 1) by 1.175, 0.829, and 0.623 standard deviations, respectively, with *p*-values = 0.005, 0.039, and 0.049, respectively. The effect size was significantly higher with the use of measurement scale Type 4 (time spent online) and measurement scale Type 3 (severity of internet addiction) than with the use of measurement scale Type 1 (anxiety) by 1.171 and 0.880 standard deviations, respectively, with both *p*-values = 0.004 after adjusting for risks of mean age and type of treatments.

Random-effects models have an underlying assumption that a distribution of effects exists, resulting in heterogeneity among study results, known as τ2, the I-squared residual, which was 90.32%. The proportion of the between-study variance explained by the meta-regression model, adjusted R-squared, was 14.65%. The corresponding meta-regression plot is shown in Figure 1.

We also tried to use the extracted data and additional results of meta-regression in Table 1, Table 2 and Table 3 to improve the goodness-of-fit indices both quantitatively (I-squared residual and adjusted R-squared) and visually (Figure 2). The results of the best fitted model are shown in Table 4, and we summarized our findings as follows: (1) the major results were consistent with those presented in Table 3; (2) after adjusting for confounding risks of age, publication year, type of comorbid diagnosis, and type of study design, the best effective intervention was pharmacotherapy combined with CBT or MLC to reduce the time spent online (Type 4) followed by lowering the severity of internet addiction (Type 3), with *p*-values = 0.006 and 0.002, respectively; (3) among three types of comorbid diagnosis (IA, IA comorbid with depression, and IA comorbid with ADHD), those youth with IA comorbid with depression (Type 2) had the worst treatment effects, after adjusting for the effects of other factors presented in the model; (4) the corresponding residual variation due to the heterogeneity in this meta-regression model, the I-squared residual, was 89.74%. The proportion of between-study variance explained by the meta-regression model, adjusted R-squared, was 16.1%. The corresponding meta-regression plot is shown in Figure 2.

### 5.3. Checking for Publication Bias

The risk of bias was assessed by funnel plots and Egger’s regression intercept tests. As shown in Figure 3, the publication bias was significant (*t* = 8.71, *p*-value < 0.001 for Egger’s test and z = 5.25, *p* < 0.001 for Begg’s test). In other words, some potential bias existed within the study.

## 6. Discussion

To our knowledge, this additional meta-regression is the largest updated meta-analysis conducted in the field of internet gaming disorder. Compared to the meta-analysis of 16 studies and 670 participants in the general population eight years ago by Winkler et al. [19], this meta-regression used 124 studies with 5601 children and young adults to contribute to the rapid development of this field.

Like the previous meta-analysis of treatment efficacy for gaming disorder in 2013, this study also found combination therapy with pharmacotherapy and cognitive behavioral therapy (CBT) or multi-level counseling (MLC) to be the best treatment type among the five types of IA/IGD treatment modalities. Either the time spent online scale or the severity of internet addiction symptom scale is a better measurement tool for studying the efficacy of treatment for internet gaming disorder. The most interesting finding is that internet gaming disorder comorbid with depression resulted in a poorer treatment effect than in patients with comorbid ADHD.

The prevalence of internet addiction has increased dramatically [35]. Especially during lockdowns due to the COVID-19 pandemic, internet gaming disorder might have a severe impact on adolescent mental health [36]. When a child or adolescent excessively uses internet gaming, they may first become unable to prioritize or keep schedules, avoid work, or procrastinate, then develop symptoms of depression or anxiety, and may even appear agitated when forced to stop using the computer by a caregiver. Therefore, gaming-addicted youth urgently need an early and effective treatment and prevention program, especially during the COVID-19 pandemic [5,37].

Internet addiction and internet gaming disorders are not just creating an important public health problem for youth. In 2016, Sepede et al. published a systematic review based on a total of 18 related articles (666 patients in total); the brains of these victims of gaming disorder showed defects in the cortical and subcortical regions of the orbitofrontal cortex, insula, anterior and posterior cingulate cortex, temporal lobe, parietal lobe, brain stem, and caudate nucleus [38]. Such defects are directly/indirectly similar to those in substance addiction and cause dysfunction in cognitive control and brain reward processing of youth. Therefore, internet gaming disorder (IGD) is a formal mental disorder in the 11th revision of the International Classification of Diseases (ICD-11), resulting in better treatment and early prevention for those children and adolescents with internet gaming disorder [39].

Parents, sociologists, psychologists, and psychiatrists worldwide have all started to seek various interventions. Fundamentally, the choice of which internet gaming disorder interventions to use shows a strong cultural bias. Generally, advanced countries with well-funded mental healthcare systems prefer evidence-based combination therapy as a first-line treatment, while some developing countries prefer to use “boot camps”, even though correction programs have been reported to be largely ineffective interventions [40]. Still, in many Asian countries, especially in China or Korea, the “boot camp”-style correction program of managing internet gaming disorder is extremely popular. Of course, this phenomenon of parent’s irrational choice of treatment reflects parents’ unwillingness to accept that internet addiction is a mental disorder, or parents’ misunderstanding that pharmacological treatment is harmful for gaming-addicted youth. Thus, one important aspect of this research is to encourage parents of gaming-disordered youth to seek a diagnosis of IGD early, along with any of its common comorbid mental disorders, e.g., ADHD and depression, and treat them with an appropriate combination intervention to prevent bad consequences in the future.

To strengthen the mental health education of youth worldwide with internet gaming disorder, governments around the world should promote appropriate and effective treatment guidelines, including how much time youth should spend online, early recognition and diagnosis of IGD, and proper simultaneous treatment of comorbid mental health disorders. The authors suggest the following as a starting point for promoting appropriate and effective treatment guidelines: we have to focus better on prevention for at-risk cases, and only provide treatment for those who have IGD. For those teenagers not yet diagnosed with internet gaming disorder, focus on training their parents, teachers, and other supporters to deal with issues when they arise early, rather than waiting for more serious problems to manifest. In addition, appropriate professional counseling for children, adolescents, and their families, or timely screening of IGD, is necessary [41]. The rule of thumb is that establishing healthy online use habits early can prevent children falling victim to internet gaming disorder. For those youth with internet gaming disorder, comorbid diagnoses should be made and treated early by a child psychiatrist. A better treatment outcome can be reached for comorbid ADHD or depression when treated properly with a combination of pharmacotherapy and psychosocial therapy.

The shortcoming of this study is that this paper pertains to internet addiction (IA) as well as internet gaming disorder (IGD). None of the intervention studies separated the treatment of internet gaming addiction and internet addiction because both internet gaming addiction and internet addiction fall under the DSM-V classification of IGD. None of the intervention studies separated the treatment effects according to treatment outcomes of internet gaming addiction and internet addiction, respectively, till now. The diagnosis of IGD in the DSM-V lumps both internet gaming disorder and internet addiction together into IGD. Because our study data was extracted from studies conducted from 2000 to 2017, IA and IGD are synonyms in this study. Therefore, we expressed behavior addiction during this mixed transient stage of nomenclature as IGD/IA. In the future, a well-designed psychometric study for evaluating the effects of treatment for IGD and IA separately is warranted. Another limitation is that the present study of effect estimates lacks long-term follow-up data to assess treatment effect durability. Finally, although we planned to identify all the studies from both Western and Eastern countries, despite a robust search strategy, most of our included papers were published by researchers from China and South Korea. This might limit worldwide-generalized conclusions based on these findings. We admit that our database research was conducted using PubMed/MEDLINE, Cochrane Library, and Airiti Library, and not the PsycInfo database, which includes more research on psychological treatments. Furthermore, we did not include more recent studies, e.g., of new transcranial direct current stimulation (tDCS), on the efficacy of treatments for IGD after 2017 [42]. Therefore, the inclusion of articles published only before 2017 is also one of the limitations of this article.

## 7. Conclusions

The value of this article is that it reinforces Winkler’s previous results. We included all the articles in either English or Chinese; all of the existing interventions, even including alternative therapies such as biofeedback, acupuncture, and electro-acupuncture; more comorbid diagnoses including ADHD, depression, and anxiety; more measurement tools; and greater confinement to children and young adults for meta-regression analysis, which makes the study results more significant.

In short, this systemic review, a meta-regression of randomized clinical trial studies, can provide child mental health professionals with effective treatment modalities for internet gaming disorder. The results of this meta-regression caused clinicians to become interested in earlier diagnosis of youth with internet gaming disorder. More psychiatrists must be invited to join in the prevention of internet gaming disorder. In addition, government policy makers for adolescent mental health should encourage more child and adolescent psychiatrists to provide pharmacotherapy combined with cognitive behavioral therapy (CBT) or multi-level counseling (MLC) to obtain the best treatment outcome.

## Figures and Tables

**Figure 1 ijerph-19-02612-f001:**
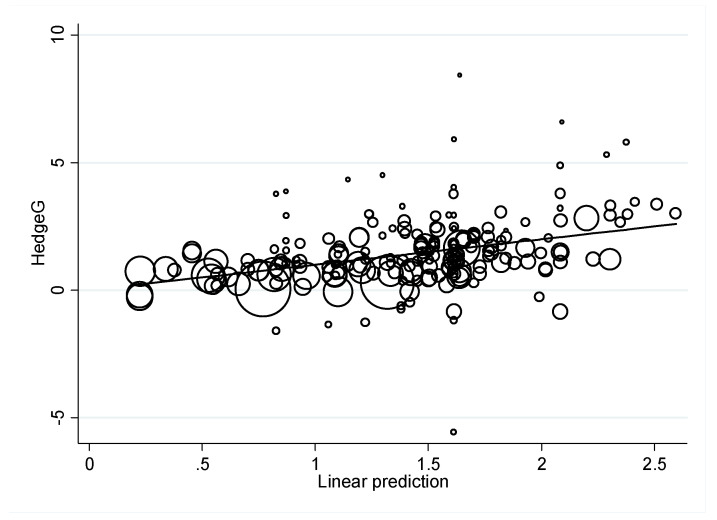
Meta-regression plot, τ^2^ = 1.209, I^2^-Residual = 90.32%, Adjusted-R^2^ = 14.65%, *n* = 207.

**Figure 2 ijerph-19-02612-f002:**
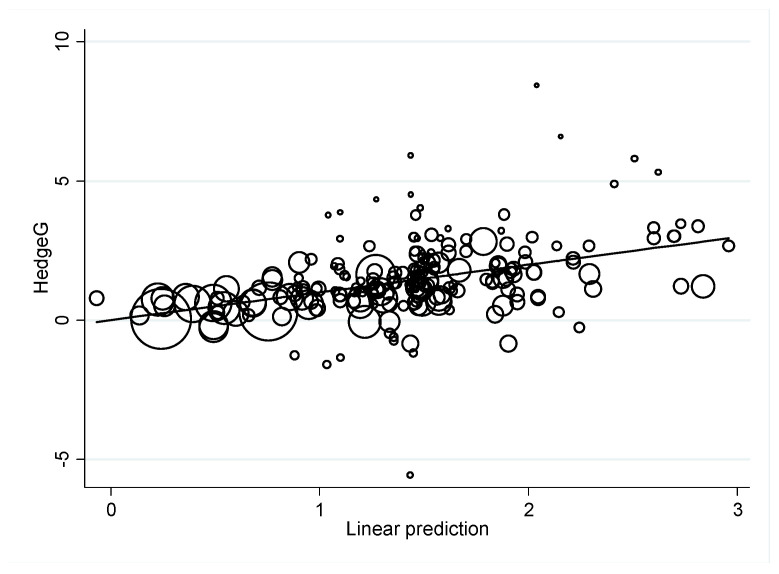
Meta-regression plot, τ^2^ = 1.188, I^2^-Residual = 89.74%, Adjusted-R^2^ = 16.10%, *n* = 207.

**Figure 3 ijerph-19-02612-f003:**
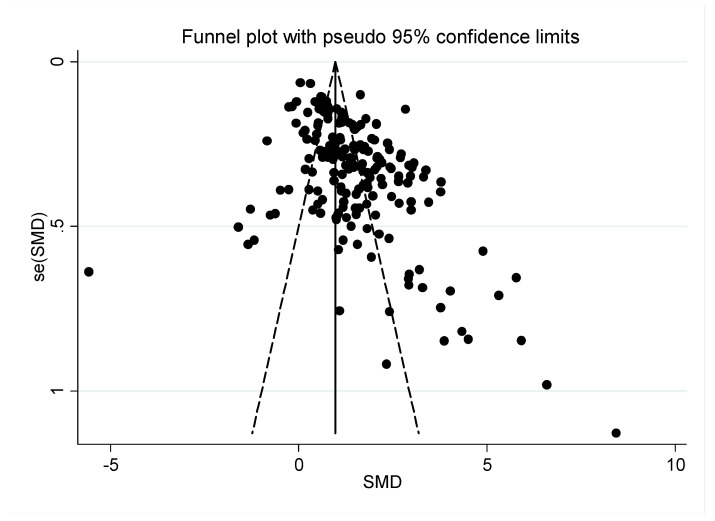
Funnel plots for treatment effects of children and adolescents with Internet addiction. The Egger’s regression intercept testing for bias was significant.

**Table 1 ijerph-19-02612-t001:** Summary of study characteristics of all included papers.

Paper	Year	Country	No. of Studies	Age	Measurements	No. of Patients	Treatment Sections	Type of Subjects	Treatment
Cao FL et al.	2007	China	1	14.8	YDQ, CIAS	57	8	IA	CBT vs. Control
Li G & Dai XY	2009	China	1	16.5	CIAS	76	12	IA	CBT vs. Control
Shao Z et al.	2015	China	1	16	SDS, SCL-90	66	8	IA + Depression	CBT + Drug vs. CBT
Wei QX	2008	China	1	17	SAS, SDS	60	12	IA	Others (Psychology Nursing) vs. Control
Yang FR et al.	2005	China	1	15.2	IAD-DQ, SCL-90, Time	52	12	IA	SFBT (No control)
Liao XC	2010	China	1	15	IAD-DQ, SCL-90	284	32	IA + Depression	CBT + Drug vs. CBT
Wu LZ et al.	2007	China	1	19.5	Time	27	6	IA	Others (HANS) vs. Control
Pan SJ et al.	2010	China	1	17	CIAS, SCL-90, SDS	11	16	IA	Others (EEG Biofeedback) (No control)
Liao YR et al.	2012	Taiwan	1	14	CIAS-R, Time	18	8	IA	Others (Adlerian Group Counseling) vs. Control
Li & Dai	2009	China	1	16.5	CIAS	76	9	IA	CBT vs. Control
Bai & Fan	2007	China	1	19	CIAS	48	8	IA	MLC vs. Control
Du YS et al.	2010	China	1	15.9	IOSRS	56	8	IA	CBT vs. Control
Han DH et al.	2009	Korea	1	9.3	YIAS-K, Time	62	8	IA + ADHD	Drug (Methylphenidate) (No control)
Yang R et al.	2005	China	1	16	CIUS, SDS, SCL-90	18	8	IA	MLC + Drug (Fluoxetine) + Others (No control)
Zhu TM et al.	2009	China	1	22.2	ISS, SAS, SDS, HAMA, HAMD	45	20, 10	IA	CBT+ acupuncture vs. CBT
Yeun YR et al.	2016	Korea	33	8~12	K-scale, YIAS, IGAS	1330	6~22	IA	CBT, MLC, CBT/MLC + Others, Others
Liu J et al.	2017	China, Korea	49	15.5~20	IA	2304		IA	CBT, MLC, Others
Winkler et al.	2013	China, Korea, US	16	14.8~23	Anx., Dep., IA, Time	454		IA	Drug, CBT, MLC, Others
Liu QX et al.	2015	China	1	15.7	APIUS, Time	46	6	IA	MLC vs. Control
Yang Y et al.	2017	China	1	21.1, 21.7	YIAS	32	10, 20	IA	CBT vs. Electro-acupuncture
Han DH et al.	2012	Korea	1	20.2	YIAS, BDI, Time	50	8	IA + Depression	Drug (Bupropion)
Park JH et al.	2016	Korea	1	17	YIAS, CDI	86	12	IA + ADHD	Drug (Methylphenidate) vs. Drug (Atomoxetine)
Mun SY et al.	2015	Korea	1	10.5	IA, Time	56	8	IA	CBT vs. Control
Lien T.-C.	2007	Taiwan	1	16.8	YDQ	20	8	IA	MLC vs. Control
Huang Z et al.	2010	China	1	21	CGAI	27	6	IA	MLC vs. Control
Khazaei et al.	2017	Iran	1	20	YIAS	48	10	IA	CBT (Positive psychology Interventions) vs. Control
Hui Li et al.	2017	China	1	22	ISS, SCL-90	112	10	IA	CBT + Others (EA) vs. CBT vs. Others (EA)
Kim SM et al.	2012	Korea	1	16	YIAS, BAI, BDI, Time	65	8	IA + Depression	CBT + Drug (Bupropion) vs. Drug (Bupropion)
Han DH et al.	2012	Korea	1	14.2	YIAS, Time	15	12	IA	MLC (No control)

RP: Routine Pharmacotherapy.

**Table 2 ijerph-19-02612-t002:** Results of univariate meta-regression analysis.

SMD	Coefficients	SE	t	*p*	τ^2^	I^2^-Residual	Adjusted-R^2^	No.
Year	0.025	0.027	0.93	0.354	1.415	93.00%	0.05%	207
Mean Age	0.062	0.025	2.51	0.013	1.36	92.52%	3.99%	207
Treatments					1.39	93.00%	1.83%	207
Treat 2 vs. Treat 1	0.613	0.320	1.92	0.056				
Treat 3 vs. Treat 1	0.263	0.311	0.85	0.398				
Treat 4 vs. Treat 1	0.834	0.427	1.96	0.052				
Treat 5 vs. Treat 1	0.665	0.408	1.63	0.104				
Treat 6 vs. Treat 1	0.269	0.377	0.71	0.477				
Measurements					1.39	91.89%	1.80%	207
Type 2 vs. Type 1	0.137	0.384	0.36	0.722				
Type 3 vs. Type 1	0.533	0.309	1.73	0.086				
Type 4 vs. Type 1	0.744	0.406	1.83	0.068				
Type 5 vs. Type 1	−0.107	0.531	−0.20	0.841				
Study Types					1.432	92.87%	−1.13%	207
StudType 2 vs StudType 1	−0.077	0.265	−0.29	0.772				
StudType 3 vs StudType 1	−0.298	0.310	−0.96	0.338				
StudType 4 vs StudType 1	−0.211	0.267	−0.79	0.429				
Types of Subjects					1.402	92.66%	1.01%	207
PType 2 vs. PType 1	−0.487	0.356	−1.37	0.172				
PType 3 vs. PType 1	−0.726	0.526	−1.38	0.169				
Group Therapy					1.420	93.01%	−0.29%	207
Yes vs. No	−0.126	0.188	−0.67	0.503				
Parents Involved in the Therapy program					1.441	93.07%	−0.02%	203
Yes vs. No	0.188	0.221	0.85	0.396				

Treatments: Treat 1: Drugs; Treat 2: CBT (Cognitive Behavioral Therapy); Treat 3: MLC (Multi-level Counseling Program); Treat 4: Drug + CBT or MLC; Treat 5: Others + CBT or MLC; Treat 6: Others; Measurements: Type 1: Anxiety; Type 2: Depression; Type 3: Internet Addiction Scale; Type 4: Online Time Spent (Hours of Internet Use); Type 5: Others; Study Types: StudType 1: Two groups with pre- and post-tests; StudType 2: One group with pre-test and post-test; StudType 3: Two groups post-test only; StudType 4: Using Hedge’s g and se(G) directly; Types of Subjects: PType 1: IA; PType 2: IA + Depression; PType 3: IA + ADHD.

**Table 3 ijerph-19-02612-t003:** Results of multiple meta-regression analysis (Using significant terms only).

SMD	Coefficients	SE	t	*p*	τ^2^	I^2^-Residual	Adjusted-R^2^	No.
					1.209	90.32%	14.65%	207
Mean Age	0.087	0.025	3.53	0.001				
Treatments								
Treat 2 vs. Treat 1	0.623	0.315	1.98	0.049				
Treat 3 vs. Treat 1	0.152	0.309	0.49	0.624				
Treat 4 vs. Treat 1	1.175	0.409	2.87	0.005				
Treat 5 vs. Treat 1	0.829	0.399	2.08	0.039				
Treat 6 vs. Treat 1	0.382	0.370	1.03	0.302				
Measurements								
Type 2 vs. Type 1	0.232	0.369	0.63	0.531				
Type 3 vs. Type 1	0.880	0.304	2.89	0.004				
Type 4 vs. Type 1	1.171	0.398	2.94	0.004				
Type 5 vs. Type 1	−0.089	0.504	−0.18	0.860				

τ^2^: The estimate of between-study variance; I^2^-Residual: Residual variation due to heterogeneity in percentage; Adjusted-R^2^: Proportion of between-study variance explained by the current model. Treatments: Treat 1: Drugs; Treat 2: CBT (Cognitive Behavioral Therapy); Treat 3: MLC (Multi-level Counseling Program); Treat 4: Drug + CBT or MLC; Treat 5: Others + CBT or MLC; Treat 6: Others; Measurements: Type 1: Anxiety; Type 2: Depression; Type 3: Internet Addiction Scale; Type 4: Online Time Spent (Hours of Internet Use); Type 5: Others.

**Table 4 ijerph-19-02612-t004:** Results of multiple meta-regression analysis (with all collected potential confounding variables).

SMD	Coefficients	SE	t	*p*	τ^2^	I^2^-Residual	Adjusted-R^2^	No.
					1.188	89.74%	16.10%	207
Mean Age	0.125	0.037	3.33	0.001				
Treatments								
Treat 2 vs. Treat 1	0.528	0.365	1.45	0.150				
Treat 3 vs. Treat 1	0.092	0.374	0.25	0.805				
Treat 4 vs. Treat 1	1.044	0.458	2.28	0.024				
Treat 5 vs. Treat 1	0.505	0.462	1.09	0.276				
Treat 6 vs. Treat 1	0.171	0.422	0.41	0.685				
Measurements								
Type 2 vs. Type 1	0.283	0.370	0.76	0.446				
Type 3 vs. Type 1	1.027	0.321	3.20	0.002				
Type 4 vs. Type 1	1.125	0.409	2.75	0.006				
Type 5 vs. Type 1	−0.318	0.514	−0.62	0.536				
Year	−0.007	0.030	−0.22	0.823				
PType 2 vs. PType 1	−0.681	0.383	−1.78	0.077				
PType 3 vs. PType 1	−0.323	0.624	−0.52	0.605				
StudType 2 vs. StudType 1	0.115	0.301	0.38	0.702				
StudType 3 vs. StudType 1	0.131	0.420	0.31	0.757				
StudType 4 vs. StudType 1	−0.553	0.280	−1.97	0.050				

Treatments: Treat 1: Drugs; Treat 2: CBT (Cognitive Behavioral Therapy); Treat 3: MLC (Multi-level Counseling Program); Treat 4: Drug +CBT or MLC; Treat 5: Others + CBT or MLC; Treat 6: Others; Measurements: Type 1: Anxiety; Type 2: Depression; Type 3: Internet Addiction Scale; Type 4: Online Time Spent (Hours of Internet Use); Type 5: Others; Study Types: StudType 1: Two groups with pre- and post-tests; StudType 2: One group with pre-test and post-test; StudType 3: Two groups post-test only; StudType 4: Using Hedge’s g and se(G) directly; Types of Subjects: PType 1: IA; PType 2: IA + Depression; PType 3: IA + ADHD.

## Data Availability

Not applicable.

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
