# Peer review of "The Comparative Efficacy of Treatments for Children and Young Adults with Internet Addiction/Internet Gaming Disorder: An Updated Meta-Analysis"

_ijerph, 2022, doi:10.3390/ijerph19052612_

Round 1

Reviewer 1 Report

Thank you for the opportunity to review this manuscript. Some of the methodology of the paper is very strong, and your work should be commended. I also have a number of concerns which should be addressed. Please see my comments below:

INTRODUCTION:

Overall you have presented a reasonable argument for this study but I do have a number of concerns which should be addressed.

First, you have not argued why there might be a difference in efficacy of various forms of treatment for IGD in children and young adults versus adults. This is key to your study and not at all described in the intro.

Second, your arguments seem to centre around updating Winkler’s 2013 analysis, but there are already several updates that I can think of (Zajac et al. 2017 and 2020; Gioia & Boursier 2019.) There are also recent reviews of CBT treatment which might be informative. You should describe how this study is different and what it can add to the discourse.

Third, I am particularly concerned that you appear to have conflated IGD with gaming disorder, both in the introduction and in the discussion. I would urge you to make clearer distinctions because diagnostically, they are separate conditions with separate clinical markers, and also because the literature is confused enough as it is.

Other miscellaneous issues:

Lines 41-43. I would question whether prolonged use leads to an increase in gaming disorder? Several studies have shown that is not necessarily the case.

Lines 49-55. Worldwide prevalence of gaming disorder has been established in several studies. Many papers also demonstrate higher rates for younger people too. Would certainly strengthen your argument.

Lines 61-63. I think you are confusing IGD (DSM-5) and gaming disorder (ICD-11). They are separate clinical conditions

Line 64. Efficacy of gaming disorder? Did you mean efficacy of treatments for gaming disorder?

Line 67. ‘Because’ does not make grammatical sense in this context.

Paragraph 102-110. I am not sure this adds much to the intro – could be condensed into one sentence and outlined in the methods or statistical analysis section.

Paragraph 111-118. This would also be better described in the methods or statistical analyses sections. In place of these final two paragraphs you should more clearly describe the aims of the review

METHOD:

Well described but I have two concerns:

Line 122. Why have you limited your research to papers up to 2017? Given there are a number of studies that have assessed treatment of IGD/IA in the last 5-years, I think you should justify your decision.

Line 150. Although I think publication year is a useful metric, it is not necessarily the case that the data was published in the same year or next. Would it have been more useful to regress the year of data collection?

RESULTS:

Results are well presented, no major concerns.

DISCUSSION:

Discussion section also needs more work. Some of my concerns:

Paragraph 321-327. While I don’t disagree that rates of IGD have increased in response to COVID, I think prevention approaches are more appropriate. More education about potential risks, more regulation around predatory mechanisms etc.

Lines 329-334. I am very sceptical of arguments around changes to brain structure in response to IGD. Just about any activity causes changes to brain structure and chemistry. While I don’t disagree that excessive amounts of gaming may be harmful during crucial developmental windows, I think that argument is a step too far.

Lines 335-338. Again, concerned about conflation between IGD and gaming disorder. IGD is a DSM-5 diagnosis, gaming disorder is an ICD-11 classification. The conditions have different symptomology and diagnostic criteria. Important that this is recognized and addressed.

Line 357. Beware of the use of the word ‘victim’.

Paragraph 351-364. Related to earlier, I don’t think there is enough discussion about prevention. Some thoughts: Inappropriate clinical diagnostic criteria and inconsistent cut-off scores lead to inflated prevalence rates of IGD/GD in the population. Some of those marginal cases are being treated through various forms of treatment, perhaps unnecessarily. Better to focus on prevention for at-risk cases, and only treatment for those who have IGD. Train parents, teachers and other support to deal with issues when they arrive early, rather than waiting for more serious problems to manifest. Prevention should also include a focus on predatory mechanisms (e.g., lootboxes and microtransactions) and how they might be contributing to IGD – especially in children and young adults.

LIMITATIONS:

Although you raise the issue of your choice of cut-off at 2017, I don’t think you have adequately explained the reasoning behind that decision. The fact that your study includes mainly Asian regions should also be discussed in more detail in the context of generalizability to Europe, USA and other parts of the world. – given prevalence rates appear to be higher in Asian regions, would we expect treatment efficacy to be higher as well? I am not sure, but worthy of discussion.

Author Response

Review 1: Comments and Suggestions for Authors

Thank you for the opportunity to review this manuscript. Some of the methodology of the paper is very strong, and your work should be commended. I also have a number of concerns which should be addressed. Please see my comments below:

INTRODUCTION:

Overall you have presented a reasonable argument for this study but I do have a number of concerns which should be addressed.

First, you have not argued why there might be a difference in efficacy of various forms of treatment for IGD in children and young adults versus adults. This is key to your study and not at all described in the intro.

Ans: Mainly because many recent clinicians found many adolescents and young people are obsessed with Internet addiction. But there are limited systemic review on the treatment efficacy of Internet addiction in adolescents and young people. If we use key word of ï½€internet addiction' and ï½€treatment efficacy' and ï½€meta-analysis' to screen the systemic review of the treatment effect of Internet addiction. We will find only the Internet addiction analysis of adults in 2013 by Winkler’s analysis. Therefore, this updated meta-analysis of Internet addiction focusing on young people is extraordinarily important study.

Second, your arguments seem to centre around updating Winkler’s 2013 analysis, but there are already several updates that I can think of (Zajac et al. 2017 and 2020; Gioia & Boursier 2019.) There are also recent reviews of CBT treatment which might be informative. You should describe how this study is different and what it can add to the discourse.

Ans: The Winkler’s 2013 analysis was a general meta-analysis for collecting effect size of various treatments intervention modal of IGD. Thanks to the reviewer for providing three studies on treatment effects of internet gaming disorder, but these three studies of treatment effects on IGD are not meta-analysis as ours. Our study is the updated meta-analysis, so called meta-regression to comprehensively compares the estimated effect of various pharmacotherapy and psychosocial intervention for IGD from randomized controlled trials (RCT) by updated meta-analysis, meta-regression. This is different study to provide exactly more effective treatment modal of IGD.

As for the reason why these three articles were not included in our updated meta-analysis. Because our meta-analysis was collecting treatment effect data of IGD from 2000 to 2017. After 2017, we worked hard to write and publish our study result. Possibly our writing ability or language expression ability is not good, so this study result remained in the state of unpublished. This is why we did not include more recent data of treatment of IGD.

Third, I am particularly concerned that you appear to have conflated IGD with gaming disorder, both in the introduction and in the discussion. I would urge you to make clearer distinctions because diagnostically, they are separate conditions with separate clinical markers, and also because the literature is confused enough as it is.

Ans: Thank you again for this important comment. As we known quite well that we will not find terms of IGD until the DSM-5 published in 2013. But this meta-regression of the treatment efficacy of IGD was studying data from 2000 to 2017. Therefore, we found phenomenon of the conflated IGD with gaming disorder. Therefore, we showed this limitation in discussion as following: “The shortcoming of this study is this paper pertains to Internet Addiction (IA) as well as Internet Gaming Disorder (IGD). None of the treatment intervention studies separated the treatment of Internet gaming addiction and Internet addition because both Internet gaming addiction and Internet addiction fall under the DSM-V classification of IGD. Because none of the treatment intervention studies separated the treatment effect respectively according to treatment outcome of Internet gaming addiction and Internet addiction till now. In diagnosis of IGD from the DSM-V, they lump both gaming and Internet addiction together into IGD. Because our study data extracted from 2000 to 2017, IA and IGD are synonyms in this study. Therefore, we expressed behavior addiction during this mixed transient stage of nomenclature as IGD/IA. In the future, a well-designed psychometric study for evaluating the treatment effect of both IGD and IA separately is warranted.”

Other miscellaneous issues:

Lines 41-43. I would question whether prolonged use leads to an increase in gaming disorder? Several studies have shown that is not necessarily the case.

Ans: Thank you for your valuable suggestion. Indeed, we know well prolonged using internet may not directly lead to an increasing in gaming disorder. This description is just for the clinician’s clinical general experience. Because many parents complain that their children playing game too much during pandemic lock-down period. Coincidentally, several studies really reported increasing rate of internet addiction or gaming disorder during COVID-19. Thank you for reviewer’s comment. We would like to add word like “Generally, clinician impressed the”

Lines 49-55. Worldwide prevalence of gaming disorder has been established in several studies. Many papers also demonstrate higher rates for younger people too. Would certainly strengthen your argument.

Ans: Thank you for your valuable comments. Here we first highlight the worldwide prevalence of gaming. Then, we highlight gaming disorder is very prevalent as high as 8% for child and adolescent. Thank you for your suggestion. Here in order to strengthen our selling point, we have added some word like ”inevitably “to highlight “Gaming disorder is rising significantly among child and adolescent population:” We also added this sentence in the end of this paragraph on introduction.

Lines 61-63. I think you are confusing IGD (DSM-5) and gaming disorder (ICD-11). They are separate clinical conditions

Ans: Thank you for reminding us not using confusing IGD (DSM-5) and gaming disorder (ICD-11). Mainly because this is a meta-analysis study. As we replied on previous question. None of the treatment intervention studies separated the treatment of Internet gaming addiction and Internet addition because both Internet gaming addiction and Internet addiction fall under the DSM-V classification of IGD. In diagnosis of IGD from the DSM-V, they lump both gaming and Internet addiction together into IGD. Because our study data extracted from 2000 to 2017, IA and IGD are synonyms in this study.

Line 64. Efficacy of gaming disorder? Did you mean efficacy of treatments for gaming disorder?

Ans: Thank you for reminding. The efficacy here the treatment effect of Internet addiction

Line 67. ‘Because’ does not make grammatical sense in this context.

Ans: Thank you for correcting us. We will delete this word.

Paragraph 102-110. I am not sure this adds much to the intro – could be condensed into one sentence and outlined in the methods or statistical analysis section.

Ans: Thanks for the suggestion. we will move this part to statistical analysis section.

Paragraph 111-118. This would also be better described in the methods or statistical analyses sections. In place of these final two paragraphs you should more clearly describe the aims of the review

 Ans: Thank you for this suggestive reminding. We have moved this part to statistical analyses sections. Therefore, these final two paragraphs might become more clearly expressed the aim of the review.

METHOD:

Well described but I have two concerns:

Line 122. Why have you limited your research to papers up to 2017? Given there are a number of studies that have assessed treatment of IGD/IA in the last 5-years, I think you should justify your decision.

Ans: This question has already been answered in the previous question. We are deeply sorry that this article has not been accepted on 2017 before due to poor English expression.

Line 150. Although I think publication year is a useful metric, it is not necessarily the case that the data was published in the same year or next. Would it have been more useful to regress the year of data collection?

Ans: Yes, this is a good point, and we agree with you. However, the year of data collection usually is not precisely presented in the paper. And the duration of data collection for all collected papers might not be comparable. Therefore, we used the paper's publication year instead just for consistency and convenience.

RESULTS:

Results are well presented, no major concerns.

DISCUSSION:

Discussion section also needs more work. Some of my concerns:

Paragraph 321-327. While I don’t disagree that rates of IGD have increased in response to COVID, I think prevention approaches are more appropriate. More education about potential risks, more regulation around predatory mechanisms etc.

Ans: Thank you for your constructive comments. We will consider to write more education about potential risks, more regulation around predatory mechanisms.

Lines 329-334. I am very sceptical of arguments around changes to brain structure in response to IGD. Just about any activity causes changes to brain structure and chemistry. While I don’t disagree that excessive amounts of gaming may be harmful during crucial developmental windows, I think that argument is a step too far.

Ans: In 2016, Sepede et al. published a systematic review based on a total of 18 related articles (666 patients in total); the brains of these victims of gaming disorder showed defects in the cortical and subcortical regions of the orbitofrontal cortex, insula, anterior and posterior cingulate cortex, temporal lobe, parietal lobe, brain stem, and caudate nucleus. This study aims to remind parents keep eye on potential risk of IGD. It is better to prevent their children overuse internet because they will have obvious emotional disorders or depressive disorder in the future.

Lines 335-338. Again, concerned about conflation between IGD and gaming disorder. IGD is a DSM-5 diagnosis, gaming disorder is an ICD-11 classification. The conditions have different symptomology and diagnostic criteria. Important that this is recognized and addressed.

Ans: We have already mentioned the same problem in the previous questions.

Line 357. Beware of the use of the word ‘victim’.

Ans: We will change this word to another word like patient.

Paragraph 351-364. Related to earlier, I don’t think there is enough discussion about prevention. Some thoughts: Inappropriate clinical diagnostic criteria and inconsistent cut-off scores lead to inflated prevalence rates of IGD/GD in the population. Some of those marginal cases are being treated through various forms of treatment, perhaps unnecessarily. Better to focus on prevention for at-risk cases, and only treatment for those who have IGD. Train parents, teachers and other support to deal with issues when they arrive early, rather than waiting for more serious problems to manifest. Prevention should also include a focus on predatory mechanisms (e.g., lootboxes and microtransactions) and how they might be contributing to IGD – especially in children and young adults.

 Ans: Thanks for the guide, we added the reviewer's comments in discussion.

LIMITATIONS:

Although you raise the issue of your choice of cut-off at 2017, I don’t think you have adequately explained the reasoning behind that decision. The fact that your study includes mainly Asian regions should also be discussed in more detail in the context of generalizability to Europe, USA and other parts of the world. – given prevalence rates appear to be higher in Asian regions, would we expect treatment efficacy to be higher as well? I am not sure, but worthy of discussion.

Ans: We have shown this weakness on limitation of discussion as following: “Originally, although we planned to identify all the studies from both Western and Eastern countries, despite a robust search strategy, our search results only found results from Asian countries, mainly Chinese and Korean studies. Thus, in this study, most of our included papers were published by researchers from China and South Korea. This might limit worldwide-generalized conclusion based on this finding.”

Reviewer 2 Report

REVIEW

I congratulate the Authors for the work done. I have several suggestions for improvement of this manuscript.

  1. ABSTRACT

Written correctly but too extensively. I propose to shorten this part.

  1. INTRODUCTION

Written correctly. However, it requires minor adjustments:

a)    the   Authors cite research but there is no reference to the literature (lines 66-68):

According to Winkler et al. (2013), effective treatment options for internet addiction are combining pharmacological therapies with psychological intervention (with Hedges’ g=1.61) through a meta-analysis of a randomized control trial” […?].

b)  lines 80-81  

The Authors give the data (about Winkler et. al research) that were already described earlier, ie in line 66. I suggest not duplicating this information.

  1. Research question and objectives

Did the Authors use any of the tools when asking questions and research goals?. Two popular tools are usually used: PICO (Population, Intervention, Comparison, Outcome - is mainly used in the quantitative synthesis of evidence) or SPIDER (Sample, Phenomenon of Interest, Design, Evaluation, Research type - was proposed as a method for qualitative and mixed methods search).

  1. MATERIAL AND METHOD

Written correctly and clearly.

  1. RESULTS

Well described and illustrated with tables and figures.

  1. DISCUSSION                                                                                                          a)            Authors wrote: “The most interesting finding is gaming disorder comorbid with depression got poorer treatment effect than patients comorbid with ADHD” (lines 318-319).

Could the authors discuss this observation?.

b) the Authors needlessly duplicate the information in line 336(about WHO) they already mentioned in the introduction part (lines 61-62).

  1. REFERENCES

Additional checking according to the journal instructions is recommended. They are not written correctly (Author 1, A.B.; Author 2, C.D. Title of the article. Abbreviated Journal Name Year, Volume, page range), for example:

Zhu, S.; Zhuang, Y.; Lee, P.; Li, J. C.; Wong, P. W. C., Leisure and Problem Gaming Behaviors Among Children and Adolescents During School Closures Caused by COVID-19 in Hong Kong: Quantitative Cross-sectional Survey Study. JMIR Serious Games 2021, 9, (2), e26808.

Author Response

Review 2: Comments and Suggestions for Authors

I congratulate the Authors for the work done. I have several suggestions for improvement of this manuscript.

  1. ABSTRACT

Written correctly but too extensively. I propose to shorten this part.

Ans: thank you for suggestion. We have shortened our abstract.

  1. INTRODUCTION

Written correctly. However, it requires minor adjustments:

Ans: thank you again. We have done minor adjustments.

  1. a)    the   Authors cite research but there is no reference to the literature (lines 66-68):

According to Winkler et al. (2013), effective treatment options for internet addiction are combining pharmacological therapies with psychological intervention (with Hedges’ g=1.61) through a meta-analysis of a randomized control trial” […?].

Ans: thank you for suggestion. we will add the reference.

  1. b)  lines 80-81  

The Authors give the data (about Winkler et. al research) that were already described earlier, ie in line 66. I suggest not duplicating this information.

Ans: thank you for suggestion, we have deleted some duplicated information on 80-81.

  1. Research question and objectives

Did the Authors use any of the tools when asking questions and research goals?. Two popular tools are usually used: PICO (Population, Intervention, Comparison, Outcome - is mainly used in the quantitative synthesis of evidence) or SPIDER (Sample, Phenomenon of Interest, Design, Evaluation, Research type - was proposed as a method for qualitative and mixed methods search).

Ans: The PICO format is commonly used in evidence-based clinical practice. This format creates a "well-built" question that identifies four concepts: (1) the Patient problem or Population, (2) the Intervention, (3) the Comparison (if there is one), and (4) the Outcome(s). thank you for suggestion. We did not use listed tool inside this article. But we have performed the PRISMA check as attached file.

  1. MATERIAL AND METHOD

Written correctly and clearly.

  1. RESULTS

Well described and illustrated with tables and figures.

  1. DISCUSSION                                                                                                          a)            Authors wrote: “The most interesting finding is gaming disorder comorbid with depression got poorer treatment effect than patients comorbid with ADHD” (lines 318-319).

Could the authors discuss this observation?.

Ans: Recent clinician found if teenagers’ depression co-occurring with internet overuse, their treatment effect of depression is poor. This study result has indicated for those internet addicted young people, their treatment effect of depression is really poor than those adolescent only co-occurring with ADHD.

  1. b) the Authors needlessly duplicate the information in line 336(about WHO) they already mentioned in the introduction part (lines 61-62).

Ans: we will delete “WHO” duplicated part.

  1. REFERENCES

Additional checking according to the journal instructions is recommended. They are not written correctly (Author 1, A.B.; Author 2, C.D. Title of the article. Abbreviated Journal Name Year, Volume, page range), for example:

Zhu, S.; Zhuang, Y.; Lee, P.; Li, J. C.; Wong, P. W. C., Leisure and Problem Gaming Behaviors Among Children and Adolescents During School Closures Caused by COVID-19 in Hong Kong: Quantitative Cross-sectional Survey Study. JMIR Serious Games 2021, 9, (2), e26808.

Ans: thank you again. We have corrected it.

Reviewer 3 Report

A major strength of the study is that it dealt with a timely topic. It would be helpful to elaborate further on how this topic is particularly relevant in the era of a global pandemic in order to appeal to a broader readership. 

Please be consistent with the age range that the meta-regression intends to focus on. The literature review section gives the impression that it is from childhood to adolescence, whereas in the actual body of the analysis, it appears that young adulthood (e.g., end of four-year college education in that part of the world) was intended. Please ensure that the literature review covers a similar age range as the meta-analysis itself. Also, please specify why the years 2000-2017 are the focus of this work, not between 2018 and 2021 (an updated meta-analysis typically focuses on new research published in the last 3-5 years).

Lastly, the supplementary materials that accompany the manuscript need a major revamp. Some columns in the Tables are left blank and some texts underneath the tables or figures could have been more succinct and formatted based on the APA guidelines. 

Author Response

Review 3: Comments and Suggestions for Authors

A major strength of the study is that it dealt with a timely topic. It would be helpful to elaborate further on how this topic is particularly relevant in the era of a global pandemic in order to appeal to a broader readership. 

Ans: thank you for your praise.

Please be consistent with the age range that the meta-regression intends to focus on. The literature review section gives the impression that it is from childhood to adolescence, whereas in the actual body of the analysis, it appears that young adulthood (e.g., end of four-year college education in that part of the world) was intended. Please ensure that the literature review covers a similar age range as the meta-analysis itself. Also, please specify why the years 2000-2017 are the focus of this work, not between 2018 and 2021 (an updated meta-analysis typically focuses on new research published in the last 3-5 years).

Ans: Because our meta-analysis was collecting data from 2000 to 2017. After 2017, we worked hard to write and publish our study result. Possibly our writing ability or language expression ability is not good, so this study result remained in the state of unpublished. This is why we did not include more recent data of treatment of IGD.

Lastly, the supplementary materials that accompany the manuscript need a major revamp. Some columns in the Tables are left blank and some texts underneath the tables or figures could have been more succinct and formatted based on the APA guidelines. 

Ans: thank you again. We will revamp our table and figure.

Reviewer 4 Report

Please complete the submission, adding tables you refer to.  I have access only to a very narrative description and it's impossible to review the paper without tables.  A PRISMA flowchart would also be recommended. 

I look forward to a more complete version of the manuscript with 4 tables and 2 figures that are referenced by the authors . 

Author Response

Review 4: Comments and Suggestions for Authors

Please complete the submission, adding tables you refer to.  I have access only to a very narrative description and it's impossible to review the paper without tables.  A PRISMA flowchart would also be recommended. 

Ans: we have performed a PRISMA flowchart. Check attached file.

I look forward to a more complete version of the manuscript with 4 tables and 2 figures that are referenced by the authors. 

Ans: thank you very much.

Round 2

Reviewer 2 Report

The authors have made all the suggested changes.

Reviewer 3 Report

The revised version addressed the major issues raised regarding the original version.

Reviewer 4 Report

As a rule, Authors submitting papers to IJERPH are responsible for manuscript composition as required. This also applies to the location of tables and figures in the text or in the appendix. I do not see this all the time, but I leave this issue for clarification with the managing editor. Thanks to the Editor I  have got them. I still have no access to PRISMA, however I believe that it will be attached. The comments regarding the citation of tables and the listing of articles in the review have been taken into account. So I have no other objections.